# Use Case: Ontologies and RDF-star for Knowledge Management

Bob Kasenchak[0000-0001-7324-2594], Ahren Lehnert[0000-0002-8691-6487], and Gene Loh[1][0000-0002-4538-4023]

[1] Synaptica, LLC, 11384 Pine Valley Drive, Franktown, CO 80116, USA

**Abstract.** Our client in this case study is a software company which develops, publishes, and distributes video games for consoles, PCs, smartphones, and tablets in both physical and digital formats. They also create educational and cultural software, cartoons, and literary, cinematographic, and television works. It owns several brands and a diversified portfolio of franchises.

The client required a centralized vocabulary management software platform to provide standardized concepts across a decentralized, global organization to find, browse, and discover enterprise content. They needed the ability to push vocabularies out to consuming systems and users while also allowing users to suggest new concepts without requiring them to log in to the taxonomy and ontology management software.

In addition to the out-of-the-box Graphite ontology management software functionality, the client required bespoke work in the system and dedicated API connectors which became part of the common code base for all versions going forward. Their requirements presented the opportunity for Synaptica to explore uses for the new specification, RDF-star (Arndt, et al., 2021), in our implementation. As a new and developing specification in RDF graph databases, the use of RDF-star is groundbreaking work for commercial enterprise ontology management systems.

**Keywords:** Ontologies, RDF-star, Knowledge Management.

## 1    Background

### 1.1    Synaptica, LLC

Synaptica, LLC provides award-winning, enterprise-class enterprise taxonomy and ontology management software tools and professional services. Our company mission is to help people organize, categorize, and discover the knowledge in their enterprise.

Synaptica's Graphite is a powerful enterprise collaboration tool for quickly designing, building, managing, and sharing taxonomies and ontologies, also known as Knowledge Organization Systems (KOS), using an intuitive graphical user interface.

This paper describes a client case study requiring custom features and integrations to use ontologies and taxonomies throughout their enterprise knowledge management (KM) information ecosystem.

## 2      The Business Challenge

The client faced challenges with outdated content which was difficult to find through browse or search due to inconsistent or missing metadata. In addition, there was no way to discover new content and users would re-create existing content because it could not be found. Taxonomy management was inefficient and included many concepts which were simply not valuable to the business.

The client wanted to be able to manage the vocabularies centrally while integrating with numerous home-grown and commercially available systems and content repositories, including the content management systems, Atlassian Confluence and Microsoft SharePoint. These vocabularies would in turn support internal knowledge management practices including the creation, tagging, and retrieval of product-specific content and user-specific suggested content to personalized home pages.

Finally, because the ontology management system would be integrated with other business applications and required users in various business roles from across the organization to access the vocabularies, more robust, group-based permissions were required.

## 3      The Business Solution

### 3.1      Ontology Management, Workflow, and Systems Integration

The client's guiding principles for controlled vocabularies include using ontologies to drive user experience such as search, browse, and discovery. Taxonomies and ontologies are used as metadata schemes by content creators to tag by topic and language. Vocabulary editors manage the taxonomies, ontologies, and the governance process.

Our client uses Graphite for ontology management including the dedicated Confluence and SharePoint connectors. Terms from specified taxonomies (knowledge models or schemes) from Graphite are available for tagging wiki pages in Confluence. Content creators working in Confluence tag pages with the appropriate scheme concepts and are allowed to suggest concepts in the same user interface. In our client's workflow, they allow users to choose the target scheme for the suggested concept.

Within Graphite, these concepts are stored in the target scheme in the "candidate" status to be reviewed by a dedicated team of ontologists. Ontologists develop the suggested concepts by adding metadata, properties, and relationships and move them to other, appropriate schemes as necessary. The ontologists use built-in workflow tools as part of their governance process to change the concept status from "candidate" to "accepted" and eventually "published" through the concept lifecycle to make them available for tagging.

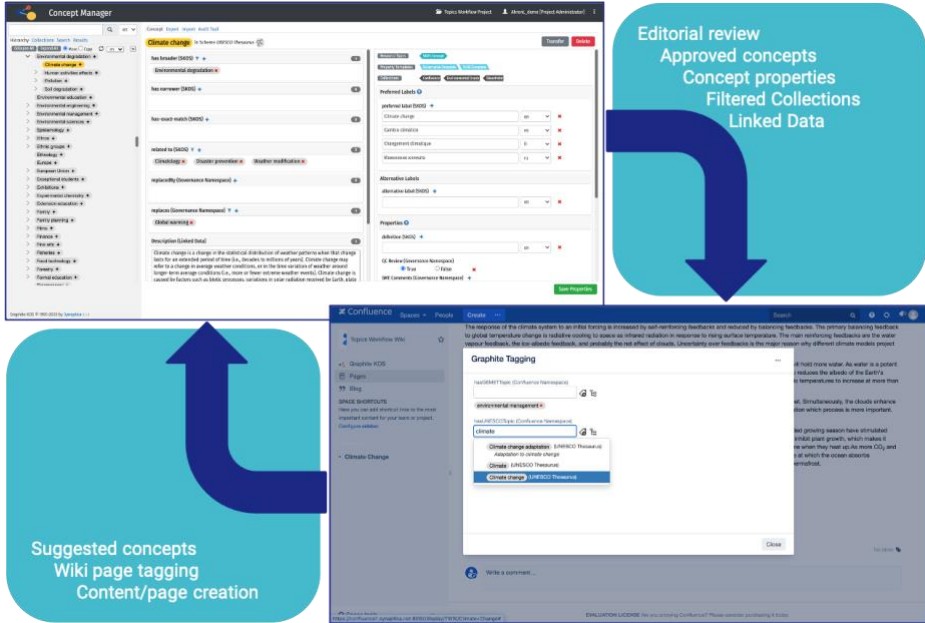

**Fig. 1.** Ontology concept workflow for wiki content tagging and new concept suggestion.

The client also required advanced concept, scheme, and project visualization, additional batch editing features, additional concept filters, and flexibility to transfer concepts across schemes. In Confluence, the ability to tag pages with existing and suggested concepts, filter available concepts by status and collections, copy and move pages and tags, and see all pages tagged with the same concepts were customized improvements to help the client realize their workflow and governance goals. Finally, specific additions to the SharePoint Connector to align schemes between Graphite and SharePoint Term Sets ensured client vocabularies could be reused across systems.

### 3.2 Permissions and RDF-star

In addition to customizations supporting knowledge management, the client required new administrative features allowing for the creation of user groups and quick assignment of users to groups with pre-selected permissions settings. Making use of new features in the back-end graph database, Ontotext's GraphDB, Synaptica was able

to deliver user and group permissions functionality on a revamped architecture utilizing RDF-star, making the creation and management of users and groups faster and with fewer triples than previously required.

## 4 The Technical Solution

### 4.1 RDF-star

RDF-star (formerly known as RDF*) helps in cases in which the user needs to express a complex relationship with metadata associated for a triple. For example,

```
1. <<:man :hasSpouse :woman>>
2.    :source :TheNationalEnquirer;
3.    :webpage <http://nationalenquirer.com/news/2020-02-12>;
4.    :retrieved "2020-02-13"^^xsd:dateTime.
```

**Fig. 2.** Complex relationship with metadata expressed in RDF.

Technically speaking, RDF-Star makes it easier to attach metadata to edges in the graph. Or, in other words, to make a statement about another statement. This was already possible in the very first RDF 1.0 specification (W3C, 1999) using the mechanism called *reification*. Unfortunately, reification introduces processing overhead due to the increased number of additional statements needed to identify the reference triple and appears too verbose when represented in RDF and SPARQL.

```
1. :man :hasSpouse :woman .
2. :id1 rdf:type rdf:Statement ;
3.     rdf:subject :man ;
4.     rdf:predicate :hasSpouse ;
5.     rdf:object :woman ;
6.    :webpage <http://nationalenquirer.com/news/2020-02-12>;
7.    :retrieved "2020-02-13"^^xsd:dateTime.
```

**Fig. 3.** Complex relationship expression in RDF using reification.

The authors of RDF-star proposed a new compact syntax. Because of its elegance, GraphDB optimized its persistence to nearly double the loading speed for datasets with a large amount of statement-level metadata. The feature immediately received interest from ontology modelers who struggled to express complex relationships in a short and concise way.

### 4.2 The Significance of RDF-star

The practical significance of RDF-star is that it increases the modeling expressivity with a new RDF resource type – an embedded triple – which works as a pointer to an

RDF statement. This also fully matches the theoretical expressivity of the property graph (PG) model without the need to use reification, i.e., an abstract construct with the existing specific methods supported by the language.

Now every PG can be efficiently represented as an RDF model. The opposite direction is not true, because RDF is more expressive in various ways. In particular, with RDF-star method can attach arbitrarily complex descriptions to an edge in the graph, while in PG one can attach only key-value pairs.

The results for a given Wikidata dataset illustrate why RDF-star is a better approach to modeling RDF statements associated with complex metadata.

| Modeling approach | Total statements | Loading time (min) | Repository image size (MB) |
|---|---|---|---|
| Standard reification | 391,652,270 | 52.4 | 36,768 |
| N-ary relations | 334,571,877 | 50.6 | 34,519 |
| Named graphs | 277,478,521 | 56 | 35,146 |
| RDF-star | 220,375,702 | 34 | 22,465 |

**Fig. 4.** RDF statements required to express relationships using various methods.

### 4.3 RDF-star and Complex Access Control Lists (ACLs)

The ACL case fits scenarios in which metadata needs to be associated with a given statement. Graphite provides users the ability to define access permissions at the property level. One of the challenges faced by our design team when attempting to extend the Graphite permissions data model pertained to the limitations of native RDF triple constructs so that extending beyond three tuples became a cumbersome exercise.

For example, "User A has edit permissions to Property B for the concepts in Scheme C in Project D", requires a tuple with a minimum of five elements for semantic expression of relationships in the dataset.

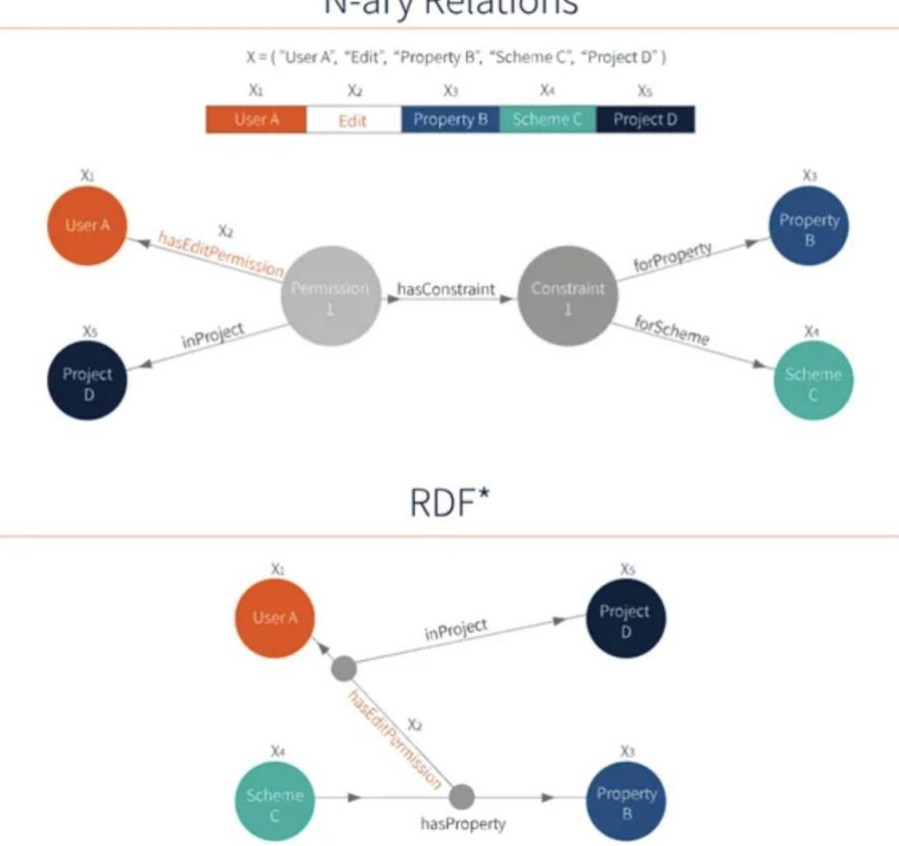

**Fig. 5.** Standard RDF relationships versus RDF-star to create permissions.

### 4.4    RDF-star and Property-Level Permissions

In the context of the Graphite permissions data model, the workaround of adding RDF triples in a conventional setting was inefficient and greatly reduced human readability of the data structures and SPARQL queries.

After consolidating the new user and group access control and permissions model in Graphite, there were no issues remapping functionalities from the old model to the new. It is a testament to the design of RDF-star and SPARQL-star that migrating existing data from the previous RDF model to RDF-star can be performed in a straightforward series of SPARQL-star statements.

In addition, the syntax of embedded triples is intuitive, which shortened the learning curve for the Graphite developers. Beyond its use in the Graphite model, RDF-star would invariably simplify the representation of ontology structures like SKOS-XL.

While the schematic representation in the data model is relatively straightforward, there are unique challenges in designing a user interface sufficiently intuitive for data entry and editing.

## 5 The Results

The use of centralized ontologies for tagging creates a unified language for the client and a foundational driving architecture for semantic applications throughout the organization. The application of controlled vocabulary concepts as metadata to content allows the client to improve browse, search, and discovery experiences on the front end within the organization. Users can browse internal content based on the metadata, see content grouped by key topics, and contribute to the ontology without accessing the ontology management software directly.

The changes to the back-end architecture and administrative functionality allow for more scalable adoption and onboarding throughout the organization as new teams begin utilizing the ontology management platform to drive semantic applications. In addition, the changes to the underlying architecture to use RDF-star makes the application faster and paves a path forward for additional functionality built on the standard.

The client's functionality requests served their immediate needs for enterprise requirements while also enhancing the semantic capabilities of the main application and connectors for all current and potential users of the product. This project reinforced Synaptica's commitment to maintaining a single software code-base and the development of specific connectors when necessary in conjunction with the use of general REST-based APIs.

## 6 Acknowledgments

Synaptica wishes to thank Vassil Momtchev, Ontotext CTO, for his contributions to this paper defining RDF-star and information detailing the Ontotext GraphDB implementation.