# OpenReview forum: "Use Case: Ontologies and RDF* for Knowledge Management"
_eswc-conferences.org/ESWC/2021/Conference/Industry_Track — ESWC 2021 Industry_

### Official Review · ~Aneta_Koleva1 · 2021-04-16
**RDF* ontology for managing vocabulary of a software company**

**Rating:** 5
**Confidence:** 4

**Review:**

This paper presents how centralized system for tagging was implemented within a software company to facilitate search.
The presentation of the idea is not very clear. The authors explain how the tool Graphite, is suitable as a solution for the needs of a specific client (software company). In the paragraph 'The Solution', the authors explain software for ontology management, Graphite, and only mention RDF* as an addition which enabled the use of user and group permission.

**Pros** - Interesting use case of how an ontology management system is used for tagging wiki pages in Confluence.

**Cons** - Not structured and presented as a conference paper. There are no references of similar work. Not clear what is the value of using RDF* star in an ontology management system.

---

### Official Review · ~Aparna_Saisree_Thuluva2 · 2021-04-19
**Paper can be considered for acceptance**

**Rating:** 7
**Confidence:** 4

**Review:**

The paper addresses the problem of knowledge management at an enterprise level and proposes a solution based on Knowledge Graphs for the problem.The approach used by the authors seems to be sound and the quality of data in the models is ensured by including the ontologists to review the concepts before adding them to the knowledge models. Overall, the papers describes nicely how knowledge graphs can be used by an enterprise in the real-world use cases in a systematic way. However, the lessons learned and limitations of the approach should be discussed in the paper, then this paper can be considered for acceptance.